# Assessment of the Impact of Increased Physical Activity on Body Mass and Adipose Tissue Reduction in Overweight and Obese Children

**DOI:** 10.3390/children10050764

**Published:** 2023-04-23

**Authors:** Marta Nowaczyk, Krystyna Cieślik, Małgorzata Waszak

**Affiliations:** 1Institute of Health and Physical Education, Jan Amos Komenski University of Applied Sciences in Leszno, 64-100 Leszno, Poland; 2Department of Biology and Anatomy, Poznan University of Physical Education, 61-871 Poznań, Poland

**Keywords:** physical activity of children, child overweight and obesity, skinfolds, Cole’s method

## Abstract

(1) Background: Within the last 30 years, growing rates of child overweight and obesity have been observed as a very concerning phenomenon in most countries worldwide. The research aim was to assess what effect increased physical activity has on reducing body mass and adipose tissue in children between 10 and 11 years of age as well as to answer the question of whether physical activity could be considered as a factor in preventing child overweight and obesity when maintaining their existing diet and lifestyle. (2) Material and methods: There were 419 sports class primary school pupils in the experimental group who, in addition to four obligatory weekly physical education hours, attended six training hours. The control group comprised 485 children from parallel non-sports classes. In all pupils, height and weight measurements as well as physical fitness measurements were taken twice, at the beginning and end of the school year. Cole’s method was used to assess the children’s normal body weight. With the help of this method, children with excessive body weight were selected from the entire study group (N = 904), who additionally had skinfolds and adipose tissue measured using the BIA electrical bioimpedance method. The obtained results were interpreted using the variance analysis for repeated measurements and LSD test. (3) Results: The number of children with excessive body mass after 10 months of increased physical activity decreased (in the case of overweight: *p* = 0.0014, obesity: *p* = 0.0022), as did their skinfolds (*p* ≤ 0.001) and body fat (*p* ≤ 0.001), while their physical fitness considerably improved compared to the control group. (4) Conclusions: The introduction of increased physical activity in the experimental group children when maintaining their existing diet and lifestyle contributed to reducing their obesity and overweight and, at the same time, proved to be an effective factor in the process of decreasing their excessive body mass.

## 1. Introduction

Called a “21st-century pandemic”, obesity is a serious health-related problem in contemporary societies and many data indicate that it is going to grow [1,2,3].

Over the last 30 years in most countries worldwide, the prevalence of overweight and obesity in the developmental age population has been observed to rise significantly at a rapid pace [4,5]. Research on the population corroborates that it is currently a health-related problem in all age groups [1,6,7,8]. What is particularly worrying, though, is the fact that the phenomenon is becoming increasingly prevalent among children and adolescents [9,10,11].

Based on research conducted in various regions of the world, it is estimated that the number of obese children and teenagers in the age group of up to 18 years has tripled [10]. The prevalence of overweight and obesity among children and teenagers between 5 and 19 years of age has grown dramatically from a mere 4% in 1975 to slightly over 18% in 2016 [12].

When analysing the Polish case, this problem is no less serious whatsoever—Polish children are putting on weight at the fastest pace in Europe, and the World Obesity Federation is signalling that obesity will have affected nearly a million children in Poland by 2030 [13,14]. For several decades a number of centres in Poland have observed an increased BMI (body mass index) and growing prevalence of overweight and obesity [15,16]. Results of the HBSC international study indicated that in Poland between 2014 and 2018 the percentage of adolescents with excessive body mass grew from 19.9% to 21.7% with a more considerable increase noted in boys than in girls [17].

Obesity is a multifaceted problem dependent on genetics, lifestyle, dietary habits, physical activity, education, and other socioeconomic factors [18,19,20,21,22,23,24,25,26,27].

A highly important environmental factor influencing obesity is adequately undertaken physical activity. According to scientists, what constitutes a key cause of child and adolescence obesity is the absence of physical activity, which is even more crucial than inappropriate nutrition [28]. More and more frequently, physical activity is seen as one of the most effective methods of preventing diseases of civilisation, to which obesity belongs [29,30,31,32].

Systematic physical activity is a powerful biological stimulus and constitutes one of main factors determining the mental and physical health of the human being. Appropriate physical activity plays a key role in shaping physical development and physical fitness in children and adolescents in universally understood mental and physical health promotion in all age groups, as well as in the prevention, treatment, and rehabilitation of a number of chronic diseases [3,33,34].

Insufficient physical activity is one of the greatest problems, as are the related challenges that inhabitants of developed countries face. Living a sedentary lifestyle, working in front of the computer, and resting in front of the TV lead to many adverse health issues. Among them, the most serious seem to be overweight and obesity, which are the main causes of hypertension, diabetes, cancer, and atherosclerosis and which therefore lead to heart attacks and strokes [35,36]. The consequences of a lack of physical activity may also be muscle atrophy, osteoporosis, decreased metabolic functions, lower metabolic rate, decreased specific and non-specific immunity of the system, poor physical performance, and an increased risk of cardiovascular diseases [37,38]. Limited physical activity also affects mental health [34,39].

Since 2020, nearly the entire world has been focusing on preventing and fighting the effects of the COVID-19 pandemic. Overweight and obesity constitute major factors escalating the risk of a severe course of COVID-19. This has been confirmed by numerous scientific studies conducted in recent months [3,40,41,42,43]. Public awareness of how important practising physical activity is in improving and/or maintaining mental and physical health has been documented in the results of the research conducted by Vancini and co-authors [44] in which they provided a copious quantity of information and video podcasts on the YouTube channel about physical exercise during the COVID-19 pandemic.

What generates special concern is the absence of physical activity among children and adolescents. Habits forged in childhood often stay with a person over their lifetime, and that is why this age group should be especially encouraged to undertake more intense physical activity [45]. An increased level of physical activity in childhood correlates to a lower risk of cardiovascular diseases, type II diabetes, and a predicted longer lifespan in the adult age [46,47].

Despite global targeted efforts to tackle childhood obesity, it remains poorly understood and challenging to manage. Currently, two common approaches to treating obesity involve decreasing energy intake through healthy eating and increasing energy expenditure with physical activity [48]. The results of the interventions used have been inconclusive or even contradictory [49,50,51,52,53]. This ambiguity in the findings of the literature is due to the lack of comparable criteria for the use of physical activity. The type of activity used, the frequency and duration of training, and the possible correlation of physical activity with other environmental factors influencing obesity (e.g., diet, sleep, sedentary lifestyle) and the size and age of the study group are important. The discrepancies in the literature regarding the effectiveness of activity against obesity have confirmed the need for studies in which physical activity is strictly defined and monitored.

Our research aimed at assessing what effect increased physical activity had on body mass and adipose tissue reduction in the 10–11-year-old children under study and answering the question of whether the physical activity factor, as one environmental factor, could be considered as counteracting obesity and overweight in children when maintaining their existing diet and lifestyle.

## 2. Material and Methods

### 2.1. Study Participants

There were 904 children aged 10 and 11 under research. Of this total, 419 sports class pupils from primary schools in Leszno, which is a town in the Wielkopolskie Province (Poland), were assigned to the experimental group (E), while the control group (C) comprised 485 pupils from parallel non-sports classes (Table 1).

The conducted research took the form of an experiment aiming to determine whether increased physical activity might be a factor in reducing body mass when maintaining the existing diet and lifestyle. The study was carried out within the framework of a health project implemented with the funds obtained from the Leszno District for the project “Prevention of posture problems and obesity among fourth-grade primary school pupils”. In conducting this experiment, the authors were inspired by the report that indicated a problem of overweight and obesity among fourth-grade children inhabiting the Leszno District.

The study excluded children who were physically disabled after sustained orthopaedic injuries and chronically ill children (genetic illnesses, cancer, metabolic disorders). The inclusion criteria were children aged 10–11 years and parents’ consent to their children participating in the study.

Chronological age classes were formed so that the class middle was a whole number, i.e., 10-year-olds were recognised as comprising the 9.50–10.49 age band and 11-year-olds 10.50–11.49 years of age. Following that, the means of analysed variables computed on the basis of measurements obtained during the calendar year were closest to expected values. Every child included in the research was assessed individually, taking account of their calendar age. In the case of Cole and co-authors’ classification [54], their specified age bands were considered.

The experimental group consisted of children from sports classes of various profiles (swimming, basketball, and football). In between the measurements, in the further part of the research (which covered one school year), the pupils were exposed to increased physical activity, i.e., in addition to the obligatory four hours of weekly physical education, they had six training hours. Training classes were organised in line with the guidelines of the Polish Swimming Federation, Polish Football Association, and Polish Basketball Association.

All the pupils (N = 904) underwent somatic measurements and fitness tests twice—at the beginning and, after 10 months had elapsed, at the end of the school year. All mandatory procedures were observed during all the measurements.

All measurements (PRE and POST) were performed by the same person, an anthropologist with 10 years of experience, who was a blind investigator (did not know whether the children belonged to the experimental or control group). Somatic measurements, i.e., height and weight, as well as fat folds and body fat levels, were taken in the morning always at the same time. Children were fasted, and after these measurements they received a light breakfast at school and were subjected to fitness measurements.

### 2.2. Ethics

Parents’ consent to their children’s voluntary, gratuitous, and fully anonymous participation in the study was obtained. Parents were also surveyed in the author’s questionnaire (Appendix A), based on which the surveyed children’s environment was assessed. The Bioethics Committee of Poznan University of Medical Sciences (decision No. 782/15) granted its approval of the study.

### 2.3. Applied Research Methods

All the pupils (N = 904) underwent measurements of body height (basis-vertex) with the use of a GMP Anthropometer to the nearest 0.1 cm. Body mass was obtained with the use of electronic scales, with accuracy to 0.1 kg. Study participants were measured when wearing sports clothes, without footwear; measurements were taken with Martin’s standard measurement method [55].

Cole and co-authors’ method was applied to assess the correctness of body mass in children [54], which proposes cut-off points relevant for age and sex to classify obesity and overweight from body mass index (BMI) international tables.

In the whole studied group (N = 904), children with excessive body mass were indicated with the use of Cole and co-authors’ method [54]. They had additional measurements taken of their skinfold. For the measurement of skinfolds, a professional HOLTAIN skinfold caliper was applied, of pressure of 10 g/mm^2^ of contact surface.

Skinfolds were measured at three strictly specified body sites, i.e., on the abdomen—a horizontal fold about 1 cm below and about 5 cm to the side of the navel; on the scapula—the fold directly below the inferior angle of the scapula; and on the arm—a horizontal fold on the back of the arm above the triceps brachii, at the middle of arm length. The result was recorded in millimetres and was the mean of two measurements.

The Eurofit Test Battery [56] and Cooper test [57] were used to determine the level of physical activity of all the studied children (N = 904). The following tests were conducted:Sit-and-reach flexibility test—reaching forward when sitting. The participant made two attempts and a better score was recorded.Standing broad jump test—long jump from the take-off line. Out of two attempts, the longer jump was recorded to the nearest centimetre.Handgrip strength test—handgrip dynamometry. The participant made two attempts with their stronger had. The better score was recorded to the nearest kilogramme.Sit-up test—trunk strength attempt. The participant made one attempt. The test result was the number of correctly performed sit-ups from lying down in 30 s.Bent-arm hang test—the functional strength test on horizontal bar. The participant performed the test only once with bare feet. The time of hang was measured to the nearest second.Agility shuttle run—10 × 5 m shuttle run. The participant performed the test once. The total time taken to complete five full repetitions (50 m in total) was measured to the nearest 0.1 s.Cooper test—endurance test. The participant performed the test once. Test participants covered as much a distance as they could in a continuous 12-min run. The score was recorded in metres.

To evaluate the environmental factors conditioning the prevalence of overweight and obesity in the population of the studied children, an author’s questionnaire was used (Appendix A). It was made certain that the children in both groups (experimental and control) were exposed to similar environmental factors regarding how they spent their leisure time, what organised forms of physical activity and family physical activity they took part in, and what kind of fast-food meals they ate.

In the experimental group (N = 419), children in whom overweight and obesity were indicated in the first measurement (N = 140) also had the adipose tissue measurement taken. The level of adipose tissue was measured by estimating body composition with the use of an AKERN BIA analyser for determining body composition by means of bioelectrical impedance analysis [58]. After adipose tissue was recorded, the result was interpreted by referring to the adipose tissue table for children, according to guidelines provided by the analyser producer. It was assumed that, in line with the manual, values above 20% for girls and above 18% for boys reflected an excessive body fat content. For girls, the set standard was 16–20%, while for boys it was 15–18%.

All measurements were taken in line with binding procedures in a room next to the swimming pool where pupils attended obligatory swimming classes. Physical fitness tests were the only activities conducted in the gymnasiums of individual schools and at the athletics stadium.

### 2.4. Statistical Methods

The collected data (related to somatic characteristics and fitness components) were included in the database in the Excel 2019–2021 software, and calculations were made in the statistical software suite Statistica 12.6 (StatSoft.Pl., Kraków, Poland).

To evaluate what impact increased physical activity had on the analysed quantitative variables, the ANOVA was applied to repeated measurements (PRE and POST). Post hoc (LSD) tests were used to determine between which groups (E-PRE, E-POST, C-PRE, C-POST) the differences for the study variables were statistically significant.

The Bonferroni correction was applied to counteract the problem of multiple comparisons. The significance level (α) was divided by the number of comparisons made (in our case—6), and the new critical *p*-value was α/6. The statistical power of the study was then calculated based on this modified *p*-value. In addition, partial eta-squared (η^2^)—the so-called effect size measurement—was calculated. Its value could be between 0–1. The obtained result η^2^ multiplied by 100% measured the variance percentage of the dependent variable explained by a given result (factor of interaction of factors). The higher the η^2^ value, the greater the variability ratio of the dependent variable that was explained by that effect.

Using the Student’s *t* test for independent variables, the size of change (PRE-POST) in the analysed traits was compared between the experimental group (E) and control group (C). In addition, percentage fractions for individual categories of BMI determined with Cole’s method were compared between PRE and POST studies.

In addition, the obtained results were interpreted in the form of various graphs.

## 3. Results

### 3.1. Effect of Increased Physical Activity on Somatic Characteristics Value and Body Composition

To determine how increased physical activity affected the values of body mass, body height, BMI, and a total of skinfolds, those values were compared in the experimental group and control group at the beginning of the school year and after 10 months, at the end of the school year. When interpreting the data of variance analysis for repeated measurements, a statistically significant difference was noted between study groups for the sum of skinfolds in both sexes and body mass of boys, and between measurements taken at the beginning and end of the school year PP (PRE, POST) for all the studied variables, as well as a noticeable influence of the effect of interaction between repeated measurements and group (PP*Group) for all variables, except for body height (Table 2). The effect of interaction meant that the indicated factors (Groups E and K and the repeated measurement) acted non-additively on the examined characteristics, showing a certain synergism.

To determine exactly which compared groups differed considerably from each other, multiple comparisons between the analysed groups were performed (E-PRE, E-POST, C-PRE, C-POST) using the LSD test. Among numerous post hoc tests, it was the least significant difference (LSD) test that revealed the highest number of significant differences. To minimise the probability of type I error, the correction to the number of conducted multiple comparisons was considered. Where six hypotheses were tested altogether, the significance level, i.e., the probability of refusing the null hypothesis when it was true, was lower and came at 0.05/6, i.e., 0.00833. This meant that for a single comparison, *p* had to be ≤0.0083 in order to state that the significance level for all comparisons was ≤0.05. Based on the obtained results presented in Table 3, Table 4, Table 5 and Table 6, statistically significant differences between pairs E-PRE and E-POST, C-PRE and C-POST, and pair E-POST and C-POST were observed regarding all the studied somatic characteristics, except for body height. Contrary to that, no significant differences were noted for any of the analysed characteristics between pair E-PRE and C-PRE.

LSD test results referring to comparisons of the studied characteristics between the distinguished groups (E-PRE, E-POST, C-PRE, C-POST) were verified with an additionally conducted Student’s *t* test for independent variables, which compared the experimental group and control group in terms of the value constituting the difference in somatic characteristic measurements recorded in study I (PRE) and study II (POST) (Table 7 and Table 8).

When juxtaposing the experimental group (E) with the control group (C) in terms of the value of difference of parameters obtained in study I (PRE) and study II (POST), no statistically significant differences were found only regarding body height; when referring to the other characteristics, these differences were significant at the level of α ≤ 0.0001 (Table 8).

The obtained η^2^ values indicated that the repeated measurement (PRE, POST) factor explained the variability of body mass in 70% and body height in 80%, while the interaction between the group and repeated measurements explained the variability of the skinfold sum in 80% (Table 2).

In addition, the size of the analysed somatic characteristics was shown graphically (Figure 1, Figure 2, Figure 3 and Figure 4) at the beginning of the school year and after the lapse of 10 months for the experimental group and control group. The analysis of the presented graphics indicated that at the beginning of the school year both studied pupil groups E and K (child sex was considered) were similar in terms of somatic characteristic values, and after the lapse of 10 months, the differences between them were substantially greater, except for body height. Similar changes to this characteristic in both groups generated the chart of line joining measurement points. Physical activity, which was increased in the experimental group for 10 months, led to differences in both studied groups, particularly when it came to the sum of skinfolds and BMI.

To determine whether the conducted experiment also showed changes in children’s body composition, the step that followed was to record the level of adipose tissue in the experimental group with the use of BIA in children with excessive body mass. It turned out that the average content of adipose tissue dropped by over 4.6% (Table 9).

A comparison was also made between sex groups in terms of the size of changes to the analysed parameters as a result of the conducted experiment, separately in the experimental group and control group (Table 10). In the former one, inter-sex differences of the examined parameters between study I (PRE) and study II (POST) were statistically significant for the sum of three skinfolds and percentage content of adipose tissue. This suggested that the values of these traits decreased in the second study more in boys than in girls (Table 10), whereas in the control group, no inter-sex differences were noted between the values of these parameters. Differences between boys and girls in study I and study II in terms of body mass, body height, and sum of three skinfolds were very comparable.

Another effect of the increased physical activity was that the parameter values of examined children moved between the adjacent BMI categories determined with Cole’s method. In both groups (experimental and control group), considerable changes related to normal weight, overweight, and obesity categories, and their direction was different. After 10 months of increased physical activity, the percentage of experimental group participants whose BMI was normal grew (from 52% to 58%), while it substantially fell in the control group (from 59% to 49%). At the same time, the percentage of children placed in the overweight and obesity category significantly decreased in the experimental group and noticeably grew in the control group (Table 11), which corroborated the impact and effectiveness of the increased physical activity factor in the process of reducing overweight and obesity in children.

To better illustrate this effect, the observed changes are presented graphically in Figure 5 and Figure 6.

### 3.2. Influence of Increased Physical Activity on Fitness Components

To specify what effect increased physical activity had on fitness component values, they were compared in the experimental and control groups at the beginning of the school year and after 10 months, at the end of the school year. When interpreting variance analysis data for repeated measurements, the statistically significant influence of these factors: group (E and C), repeated measurement (RM-(PRE) and (POST)), and their interaction (RM*Group), on all studied fitness variables (Table 12) was noticed.

The conducted LSD test for fitness components, which juxtaposed the significance of differences among distinguished groups E-PRE, C-PRE, E-POST, and C-POST, led to the conclusion that there was noticeable variability in pairs E-PRE and E-POST and E-POST and C-POST with regard to all the analysed components, and in pair E-PRE and C-PRE only when it came to the handgrip measurement, the handgrip strength test in boys, whereas there was no significant variability in pair C-PRE and C-POST with regard to the distance jump test (the standing broad jump) and the functional strength test (bent-arm hang) in boys and girls. It can be therefore stated that the experimental group and control group varied at the beginning of the school year only in terms of the value of handgrip strength measurement in boys. After 10 months, the values of all the tested fitness components substantially increased in the experimental group, while no noticeable changes in the scoring of the long jump and bent-arm hang were observed in the control group.

LSD test results were additionally verified with the Student’s *t* test for independent variables, which compared the values of the differences in the parameters obtained in study I (PRE) and study II (POST) (Table 13 and Table 14) between the experimental and control groups.

In addition, the record of examined fitness components was presented graphically (Figure 7, Figure 8, Figure 9, Figure 10, Figure 11, Figure 12 and Figure 13) at the beginning of the school year and after 10 months for the experimental group and control group.

## 4. Discussion

The results obtained at the end of the school year of the experiment confirmed the impact and effectiveness of the increased physical activity factor in the process of reducing overweight and obesity in children. The elevated level of physical activity of the examined children, when maintaining their previous lifestyle, contributed to decreasing their overweight and obesity. The prevalence rates of these two phenomena in the experimental group and control group were comparable at the beginning of the school year, yet at the end of the school year, there were noticeable differences between the groups. The percentage of children with excessive body mass fell after one year of attending the class with the increased number of physical education lessons (10 h weekly), as did their skinfolds and adipose tissue level, while their physical activity was at a considerably higher level.

Similar results were achieved by Charzewska and co-authors [59]. Their research proved that obesity in sports schools (with an average weekly number of physical education lessons of 12) did not occur in girls at all and in boys its prevalence was 0.8%, which formed a statistically significant difference when compared with how prevalent obesity was in non-sports schools (with an average number of physical education lessons of three and a half), which was 3.0% in girls and 4.2% in boys. Furthermore, obesity in non-sports schools was observed to be much more prevalent than in sports schools, although no differences were found in the preliminary analysis before children started attending sports classes. The obtained results allowed the authors to conclude that increased physical activity should be recommended when compared to the current lifestyle, to fulfil its preventive role in the development of young age obesity.

The results of our research were consistent with the study conducted by Ługowska and co-authors [60], the aim of which was to assess what influence physical activity (PA) in school exerted on the body mass of 10–12-year-old children during a two-year observation. The authors reported that during the study the percentage of excessive body mass (overweight and obesity) increased by one-fourth in children with standard PA in school (four hours per week), while it slightly decreased in those with elevated PA (10 h per week). The body mass of children after two years of increased PA became more correct than of those with standard PA. It was therefore shown that the elevated level of PA in school had a beneficial effect on children’s body mass.

Likewise, Rutkowski and co-authors [61] corroborated the effectiveness of a 10-week karate training regarding the body composition and physical fitness of younger school-age children with abnormal body mass. In total, 593 primary school children from grades 1–3 were studied. The conducted intervention revealed a statistically significant reduction in the percentage content of adipose tissue when compared to the first study (from before the intervention) as well as improved physical fitness.

In addition, the results of research carried out by Baran [62] indicated that a physical activity level affects the prevalence of overweight and obesity in children. Among 1300 examined children, 9.4% were overweight and 6.5% were obese. This group was characterised by a substantially greater share of sedentary time (*p* = 0.037 *) and a smaller share of time spent on intense physical activity (*p* = 0.037 *). In children spending <60 min per day on activity ranging from moderate to intense, the risk of overweight or obesity was higher than in children whose physical activity was >60 min per day.

On the other hand, research conducted by Januszek–Trzciąkowska and co-authors [63] did not show any statistically significant connection between physical activity level and obesity in 7–9-year-olds; nonetheless, the authors noted that in the group of boys who were physically active two to three times a week the percentage of those obese was higher than in the group of boys who were active every day (OR = 1.94; 95% reliability range: 0.92–4.1; *p* = 0.08). In the course of their study among 10–12-year-old children, Czajka and co-authors [64] found an insignificant relationship between BMI value and physical activity, which was defined as physical exertion lasting at least 20 min a day. In a cross-sectional study performed in Great Britain among children between 6 and 8 years of age in which children’s physical activity was measured with the use of an accelerometer, it was found that a higher level of physical activity among boys was connected to a lower risk of overweight and obesity (OR = 0.2; 95% reliability range: 0.04–0.88), yet no similar relationship was observed in girls [65]. However, Olson and co-authors [66] revealed at the level of statistical tendency that there was association between the number of hours spent on active play or sport and the risk of overweight and obesity in children aged 6–11 (*p* = 0.06). Entirely opposing results were published by Koca and co-authors [67], who observed higher BMI values in the group of physically active children. However, Janssen and LeBlanc [68], who in their systematic review analysed 31 observational studies concerning physical activity and excessive body mass among children, found that most research pointed to a poor or moderate connection between an elevated level of physical activity and a lower risk of overweight and obesity.

Research conducted by Bilińska and Kryst [51] aimed to determine the effects on overweight/obesity prevalence of the primary-school-based intervention programme. After one year of additional physical activities and engagement in a health-oriented education programme, the risk of becoming overweight/obese was not reduced in children (7–11 years) in the experimental group. This study indicated that such programmes require more intense interventions, possibly over a longer period of time.

The results of a review of childhood obesity control strategies based on 105 scientific articles suggested that school-based programmes can have long-term effects in a large target group. This may be due to the fact that children spend a significant part of their time at school and adopt some aspects of lifestyle there. A comprehensive intervention that includes diet and exercise appeared to be more practical. However, various projects produced sometimes controversial results. The authors believed that among different types of interventional programmes a multidisciplinary approach in schools involving the child’s family may be the best and most sustainable approach to treating childhood obesity [69].

Recent studies conducted during the COVID-19 pandemic by Makaraci and co-authors [70] confirmed the positive impact of PA (physical activity) and nutritional education programmes applied to twelfth-grade students on body mass index (BMI), PA levels, and eating habits.

In the newest literature, views on solving the problem of overweight and fighting obesity signalled the strong relationship of different interacting factors. The aim of numerous studies was to explore the independent connections between many types of behaviour and lifestyle (physical activity, MVPA—Moderate to Vigorous Physical Activity, sleep, time spent in front of the screen, and diet) and overweight and obesity in children [71,72,73,74]. General 24 h guidelines on movement were defined as 9–11 h of sleep, ≤2 h per day in front of the screen, and at least 60 min per day of MVPA. Results of cross-sectional studies of 28.048 children aged 6–17 from the China National Nutrition and Health Surveillance demonstrated that fulfilling all the guidelines on physical activity, sleep, and diet resulted in a lower BMI (all *p* < 0.01) and lower probability of overweight/obesity (all *p* < 0.05) [74]. Similar research was carried out by Roman-Viñas and co-authors [72] in which they assessed the relationship between the above guidelines and obesity in 12 countries participating in the International Study of Childhood Obesity, Lifestyle, and the Environment (ISCOLE). The obtained results revealed that whenever the analysis considered a MVPA guideline, the product of obesity possibilities was lower. Wilkie and co-authors [71] also claimed that MVPA together with a longer night’s sleep were connected to a lower risk of overweight and obesity in children aged 9–11 in Great Britain.

The European Childhood Obesity Group and European Paediatrics Academy pointes to physical activity as a pivotal factor in preventing excessive body mass in the paediatric population. In many children increased physical activity might suffice to prevent obesity. Children with correct body mass who are physically active usually have less adipose tissue than their peers who do not undertake any physical activity. Therefore, a key issue in maintaining correct body mass is to adopt an adequate level of physical activity [31].

Currently, Polish primary school pupils in grades 4–6 have four mandatory lesson hours of physical education. It should be noted that research conducted by J. Raczek as early as in the 1980s [75] demonstrated that the physical exertion requirements of a typical physical education lesson are in most cases too low and do not reach a threshold of effective physiological impact. Considering the needs of a developing child’s adaptation mechanisms, physical education lessons conducted in such a way are ineffective. Four 45 min lessons of low intensity are insufficient for children at the developmental age. It is necessary that school-age children participate in additional physical activity. A particularly helpful solution is when children at the developmental age attend sports classes in which the number of physical education lessons is increased in line with the guidelines of official associations of a given sports discipline.

What raises concerns is the fact that, during the COVID-19 pandemic, the level of school-age children’s physical activity has fallen. Pluta and co-authors [76] emphasised that in order to prevent a further decrease in adolescents’ physical activity a supportive environment of physical activity should be created for this population group. According to the authors, changes should be introduced into school physical education curricula. They should address physical education teachers and their role in modelling behaviours related to physical activity among teenagers.

### Study Limitations

The sample size limited the analysis in determining various factors regarding physical activity.

Although all primary schools (there were 10) in the city of Leszno (population 60,000) were included in the study, surveying a larger number of children would have provided greater opportunities to analyse the various factors that are components of physical activity (type of activity, training time units) and to obtain more detailed results. A larger study sample size would allow for the categorisation of children in terms of quantity and quality of physical activity practice. Future studies could be supplemented with additional age groups of children.

## 5. Conclusions

The results obtained at the end of the year proved the impact and effectiveness of the increased physical activity factor (six additional hours of physical education a week) in the process of reducing overweight and obesity in children. The percentage of children with excessive body mass after one year of increased physical activity dropped, as did their skinfolds and level of adipose tissue, while their physical fitness was at a much higher level when compared with the control group.

The results of the conducted study led to the following conclusion: introducing increased physical activity in the amount of six additional hours of physical education a week over the period of 10 months in the experimental group of the examined children, while maintaining their previous lifestyle, contributed to decreasing their overweight and obesity and, at the same time, proved to be an effective factor in the process of reducing excessive body mass in the children.

The research showed that physical activity is one of the most effective methods of preventing obesity and corroborated the need to increase the physical activity of children. The study results indicated a connection between physical activity level and obesity development, and they form an argument for preventing obesity among children. It should be acknowledged today that counteracting obesity constitutes a major social problem and an important task for physical education, which should focus its curricula on present and future health-related needs. As transpired from our study, the introduction of only the factor of increased physical activity (which, in our case, was an additional continuous six hours per week of physical education in the whole school year) was a minimal but effective way of preventing child obesity.

## Figures and Tables

**Figure 1 children-10-00764-f001:**
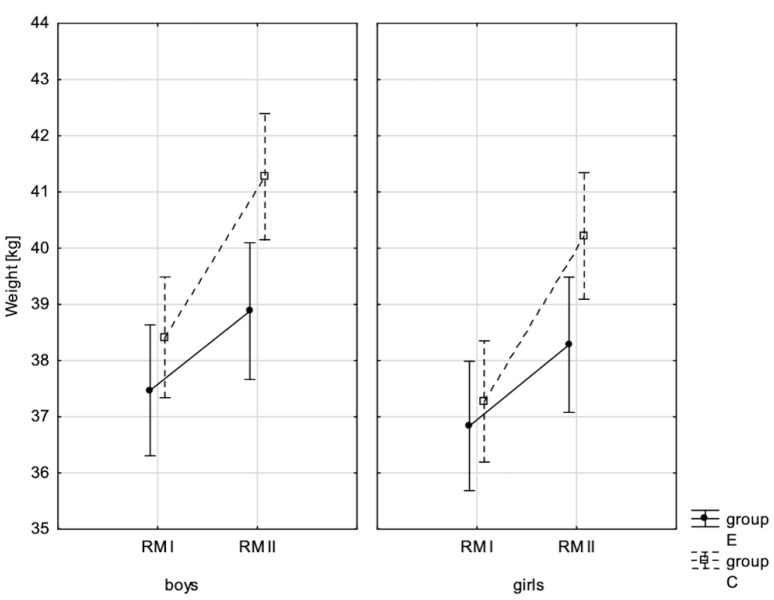
Mean values of boys’ and girls’ body weight at the beginning of the school year (RMI) and at the end of the school year (RMII) in the experimental group (E) and the control group (C). Vertical bars indicate the 95% confidence interval for the mean.

**Figure 2 children-10-00764-f002:**
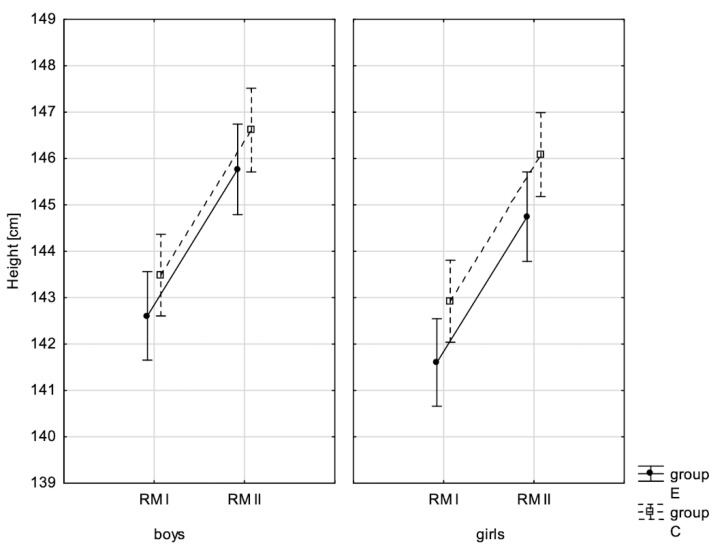
Mean values of body height for boys and girls at the beginning of the school year (RMI) and at the end of the school year (RMII) in the experimental group (E) and the control group (C). Vertical bars indicate the 95% confidence interval for the mean.

**Figure 3 children-10-00764-f003:**
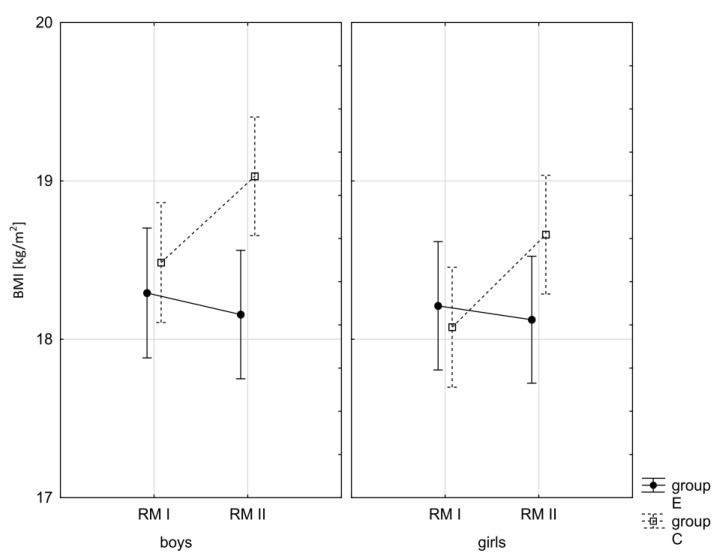
Mean values of the BMI of boys and girls at the beginning of the school year (RMI) and at the end of the school year (RMII) in the experimental group (E) and the control group (C). Vertical bars indicate the 95% confidence interval for the mean.

**Figure 4 children-10-00764-f004:**
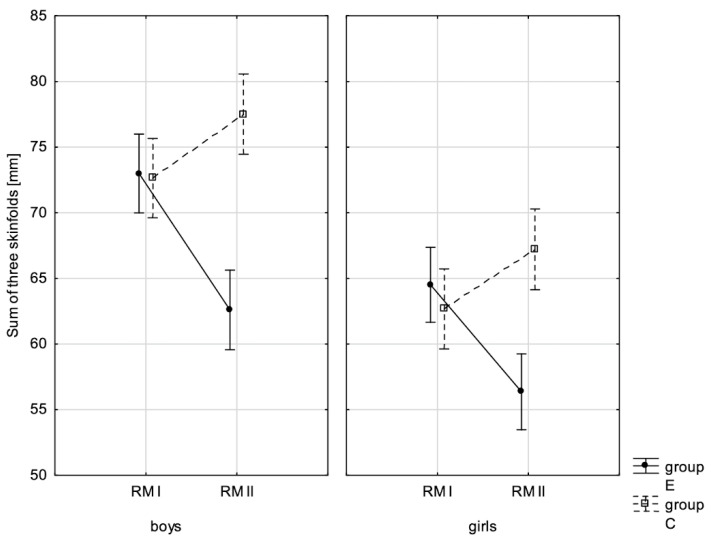
Mean values of the sum of three skinfolds of boys and girls at the beginning of the school year (RMI) and at the end of the school year (RMII) in the experimental group (E) and in the control group (C). Vertical bars indicate the 95% confidence interval for the mean.

**Figure 5 children-10-00764-f005:**
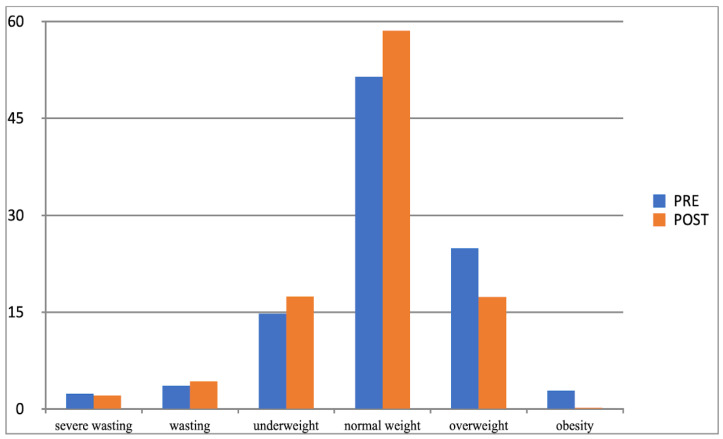
Comparison of the frequency (%) of the occurrence of categories of BMI determined by Cole’s method in the experimental group in the first and second studies (BMI-I, BMI-II).

**Figure 6 children-10-00764-f006:**
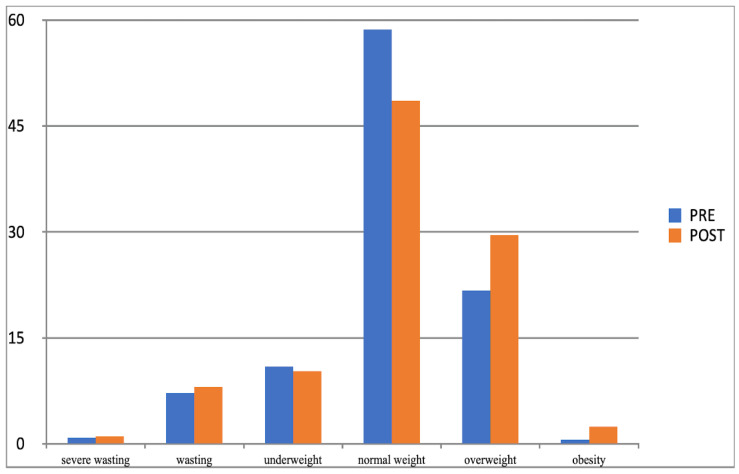
Comparison of the frequency (%) of occurrence of categories of BMI determined by Cole’s method for the control group in the I-PRE and II-POST study (BMI-I, BMI-II).

**Figure 7 children-10-00764-f007:**
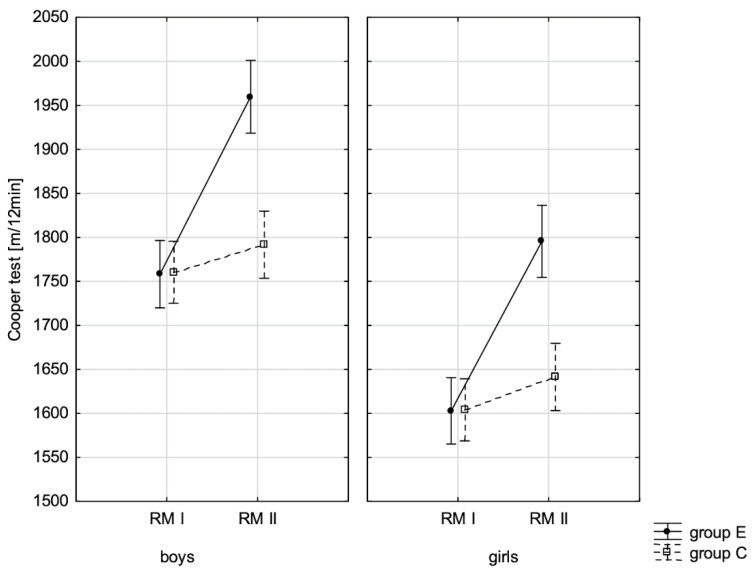
Average values of the Cooper test for boys and girls at the beginning of the school year (RMI) and at the end of the school year (RMII) in the experimental group (E) and the control group (C). Vertical bars indicate the 95% confidence interval for the mean.

**Figure 8 children-10-00764-f008:**
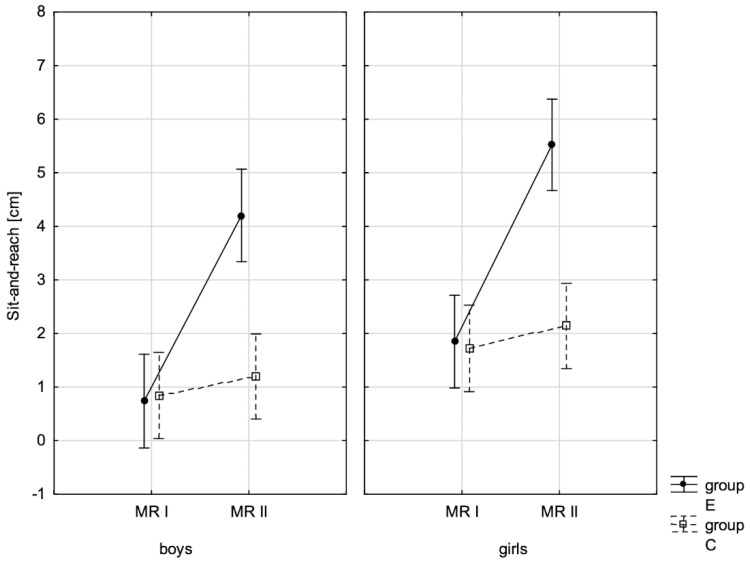
The mean values of the measurement of sit-and-reach for boys and girls at the beginning of the school year (RMI) and at the end of the school year (RMII) in the experimental group (E) and the control group (C). Vertical bars indicate the 95% confidence interval for the mean. Vertical bars indicate the 95% confidence interval for the mean.

**Figure 9 children-10-00764-f009:**
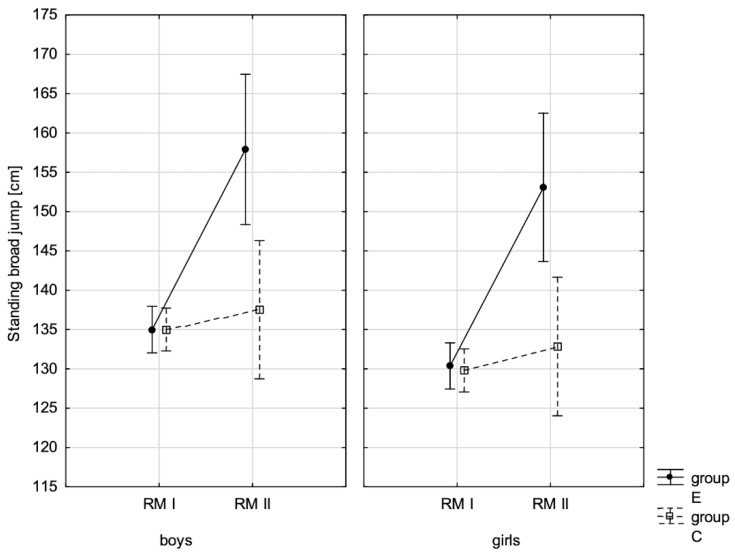
The mean standing broad jump values for boys and girls at the beginning of the school year (RMI) and at the end of the school year (RMII) in the experimental group (E) and the control group (C). Vertical bars indicate the 95% confidence interval for the mean.

**Figure 10 children-10-00764-f010:**
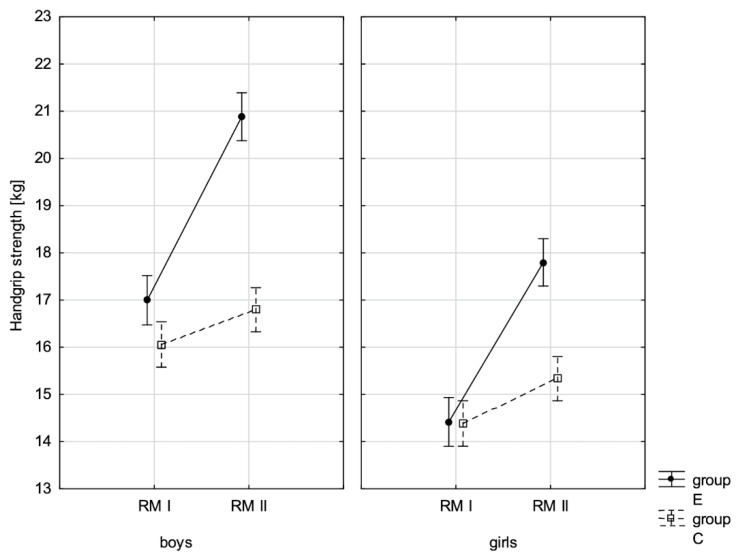
Mean handgrip strength values for boys and girls at the beginning of the school year (RMI) and at the end of the school year (RMII) in the experimental group (E) and the control group (C). Vertical bars indicate the 95% confidence interval for the mean. Vertical bars indicate the 95% confidence interval for the mean.

**Figure 11 children-10-00764-f011:**
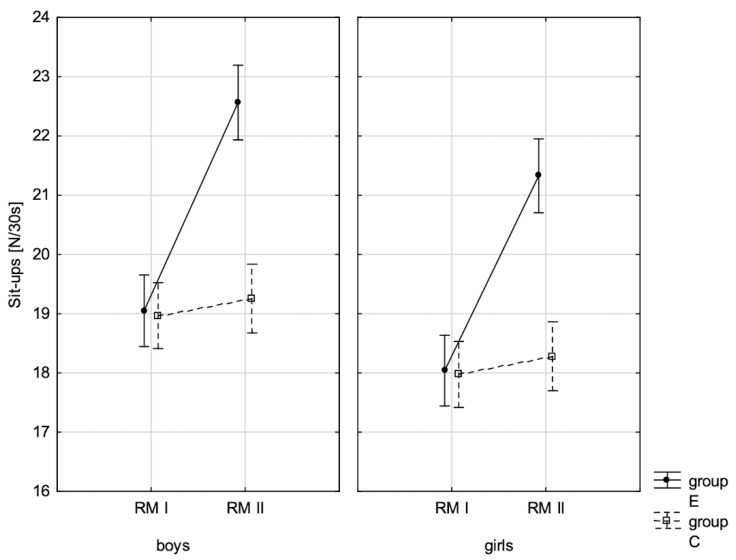
The mean values of the measurement of sit-ups in boys and girls at the beginning of the school year (RMI) and at the end of the school year (RMII) in the experimental group (E) and in the control group (C). Vertical bars indicate the 95% confidence interval for the mean.

**Figure 12 children-10-00764-f012:**
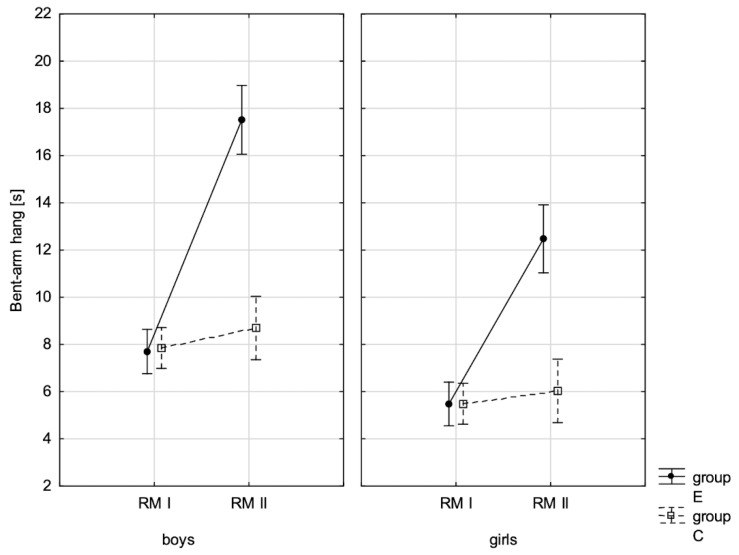
Mean values of the measurement of the bent-arm hang in boys and girls at the beginning of the school year (RMI) and at the end of the school year (RMII) in the experimental group (E) and in the control group (C). Vertical bars indicate the 95% confidence interval for the mean.

**Figure 13 children-10-00764-f013:**
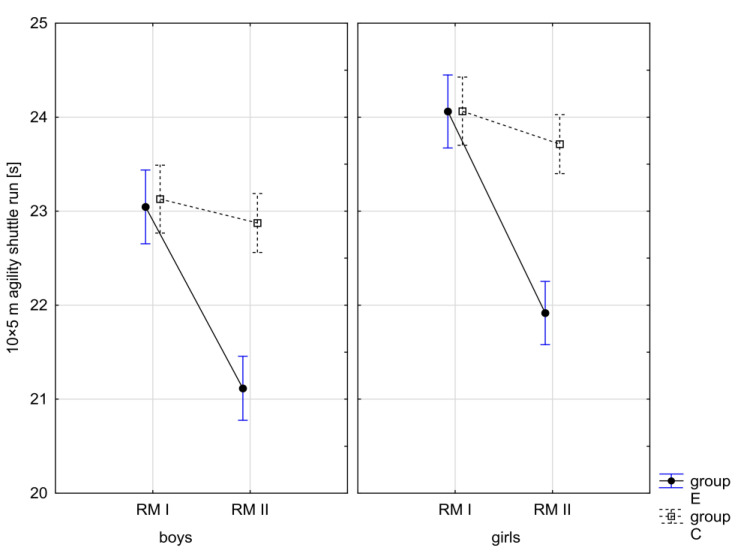
Mean values of the agility shuttle run measurement in boys and girls at the beginning of the school year (RMI) and at the end of the school year (RMII) in the experimental group (E) and in the control group (C). Vertical bars indicate the 95% confidence interval for the mean.

**Table 1 children-10-00764-t001:** Study participants.

	Experimental Group	Control Group	Total
Boys (♂)	207	243	450
Girls (♀)	212	242	454
Total (♂♀)	419	485	904

**Table 2 children-10-00764-t002:** Summary of the results of one-dimensional ANOVA with repeated measurements for somatic measurements separately for boys and girls.

Variable	Group (E, C)		RM (PRE, POST)		RM*Group	
	*p*	η^2^	*p*	η^2^	*p*	η^2^
Body weight ♂	0.049 *	0.009	0.000 **	0.709	0.000 **	0.220
Body weight ♀	0.137	0.005	0.000 **	0.704	0.000 **	0.217
Body height ♂	0.170	0.004	0.000 **	0.799	0.827	0.000
Body height ♀	0.057	0.008	0.000 **	0.821	0.894	0.000
BMI ♂	0.072	0.007	0.000 **	0.071	0.000 **	0.175
BMI ♀	0.447	0.001	0.000 **	0.135	0.000 **	0.222
Sum 3-SF ♂	0.001 **	0.076	0.000 **	0.376	0.000 **	0.822
Sum 3-SF ♀	0.029 *	0.034	0.000 **	0.222	0.000 **	0.778

♂: boys; ♀: girls; Sum 3-SF: sum of three skinfolds; E: experimental group; C: control group; RM (PRE, POST): repeated measurement—measurement at the beginning and end of the school year; RM*Group: interaction between measurements (PRE and POST) and factor Group (E, C); η^2^: partial eta-square; *p*: probability of test statistics, 0 < *p* < 0.001; ** differences significant at α ≤ 0.01; * differences significant at α ≤ 0.05.

**Table 3 children-10-00764-t003:** Significance of differences between individual groups of boys: experimental (E) and control (C) in study I (PRE) and study II (POST) for the following variables: body weight and height—LSD test results.

		Body Weight (kg)			
		E-PRE	E-POST	C-PRE	C-POST
Body Height (cm)	E-PRE		0.000 **	0.268	0.000 **
	E-POST	0.000 **		0.582	0.005 *
	C-PRE	0.165	0.0003 **		0.000 **
	C-POST	0.000 **	0.182	0.000 **	

E-PRE: experimental group, measurement in the first study; E-POST: experimental group, measurement in the second study; C-PRE: control group, measurement in the first study; C-POST: control group, measurement in the second study; *p*: probability of test statistics, 0 < *p* < 0.001; ** significant differences at α ≤ 0.01 after the Bonferroni correction; * significant differences at α ≤ 0.05 after the Bonferroni correction.

**Table 4 children-10-00764-t004:** Significance of differences between individual groups of boys: experimental (E) and control (C) in study I (PRE) and study II (POST) for the following variables: BMI and sum of three skinfolds—LSD test results.

		BMI (kg/m^2^)			
		E-PRE	E-POST	C-PRE	C-POST
Sum of three skinfolds (mm)	E-PRE		0.008 *	0.518	0.013
	E-POST	0.000 **		0.269	0.003 *
	C-PRE	0.874	0.000 **		0.000 **
	C-POST	0.046	0.000 **	0.000 **	

E-PRE: experimental group, measurement in the first study; E-POST: experimental group, measurement in the second study; C-PRE: control group, measurement in the first study; C-POST: control group, measurement in the second study; *p*: probability of test statistics, 0 < *p* < 0.001; ** significant differences at α ≤ 0.01 after the Bonferroni correction; * significant differences at α ≤ 0.05 after the Bonferroni correction.

**Table 5 children-10-00764-t005:** Significance of differences between individual groups of girls: experimental (E) and control (C) in study I (PRE) and study II (POST) for the following variables: body weight and height—LSD test results.

		Body Weight (kg)			
		E-PRE	E-POST	C-PRE	C-POST
Body Height (cm)	E-PRE		0.000 **	0.585	0.000 **
	E-POST	0.000 **		0.207	0.016
	C-PRE	0.06	0.010 **		0.000 **
	C-POST	0.000 **	0.056	0.000 **	

E-PRE: experimental group, measurement in the first study; E-POST: experimental group, measurement in the second study; C-PRE: control group, measurement in the first study; C-POST: control group, measurement in the second study; *p*: probability of test statistics, 0 < *p* < 0.001; ** significant differences at α ≤ 0.01 after the Bonferroni correction.

**Table 6 children-10-00764-t006:** Significance of differences between individual groups of girls: experimental (E) and control (C) in study I (PRE) and study II (POST) for the following variables: BMI and sum of three skinfolds—LSD test results.

		BMI (kg/m^2^)			
		E-PRE	E-POST	C-PRE	C-POST
Sum of three skinfolds (mm)	E-PRE		0.044	0.614	0.092
	E-POST	0.000 **		0.86	0.044
	C-PRE	0.375	0.003 *		0.000 **
	C-POST	0.193	0.000 **	0.000 **	

E-PRE: experimental group, measurement in the first study; E-POST: experimental group, measurement in the second study; C-PRE: control group, measurement in the first study; C-POST: control group, measurement in the second study; *p*: probability of test statistics, 0 < *p* < 0.001; ** significant differences at α ≤ 0.01 after the Bonferroni correction; * significant differences at α ≤ 0.05 after the Bonferroni correction.

**Table 7 children-10-00764-t007:** Summary of the differences in the values of parameters obtained in study I (PRE) and study II (POST) for the experimental (E) and control (C) groups.

Group		N	Body Weight I–II		Body Height I–II		BMI I–II		Sum of Three Skinfolds I–II		
			X	SD	X	SD	X	SD	N	X	SD
E	♂♀	419	−1.43	1.34	−3.15	1.65	0.11	0.78	141	9.22	4.29
	♂	207	−1.41	1.27	−3.16	1.74	0.14	0.82	67	10.40	4.23
	♀	212	−1.45	1.40	−3.14	1.55	0.09	0.75	74	8.15	4.08
C	♂♀	485	−2.90	1.44	−3.14	1.41	−0.56	0.59	131	−4.71	2.49
	♂	243	−2.86	1.44	−3.13	1.42	−0.54	0.66	66	−4.88	2.77
	♀	242	−2.95	1.44	−3.16	1.40	−0.58	0.50	65	−4.54	2.40

N: number; SD: standard deviation; X: difference of means; E: experimental group; C: control group; ♂: boys; ♀: girls.

**Table 8 children-10-00764-t008:** Intergroup comparison of the variables of the difference in measurements obtained in studies I and II (PRE and POST)—testing with Student’s *t* test for independent variables.

Variable	♂♀	♂	♀
	*p*	*p*	*p*
Body weight	0.000 ***	0.000 ***	0.000 ***
Body height	0.945	0.827	0.894
BMI	0.000 ***	0.000 ***	0.000 ***
Sum of three skinfolds	0.000 ***	0.000 ***	0.000 ***

♂: boys; ♀: girls; *p*: probability of test statistics, 0 < *p* < 0.001; *** significant differences at α ≤ 0.001.

**Table 9 children-10-00764-t009:** Comparison of the mean percentages of adipose tissue in the examined children in the first (I) and second (II) study.

Body Fat Percentage			
	Total ♂♀	Boys ♂	Girls ♀
Study I (PRE)	28.62	28.97	28.31
Study II (POST)	23.98	23.89	24.06
*p*	0.000 **	0.000 **	0.000 **

♂: boys; ♀: girls; *p*: probability of test statistics, 0 < *p* < 0.001; ** significance at α ≤ 0.01.

**Table 10 children-10-00764-t010:** Intersex comparison of differences in the values of parameters obtained in the first and second study, separately for the experimental and control groups.

Variable	Boys ♂	Girls ♀	*p*
Experimental group (E)			
Body weight I–II	−1.41	−1.45	0.776
Body height I–II	−3.16	−3.14	0.923
BMI I–II	0.14	0.09	0.524
Sum 3-SF I–II	10.40	8.15	0.002 **
BFP I–II	5.08	4.25	0.014 *
Control group (C)			
Body weight I–II	−2.19	−2.25	0.607
Body height I–II	−3.14	−3.15	0.906
BMI I–II	−0.23	−0.27	0.439
Sum 3-SF I–II	2.82	2.22	0.529

♂: boys; ♀: girls; I–II: difference between the first and second study (PRE–POST); sum 3-SF: sum of three skinfolds; BFP: body fat percentage; *p*: probability of test statistics; ** significance at α ≤ 0.01; * significance at α ≤ 0.05.

**Table 11 children-10-00764-t011:** Significance of differences between study I (PRE) and study II (POST), percentage fractions for individual categories of BMI determined with Cole’s method.

Categories		Group E			Group C		
		PRE	POST	*p*	PRE	POST	*p*
		%	%		%	%	
Severe wasting	♂♀	2.39	2.15	0.816	0.82	1.03	0.733
	♂	3.86	2.42	0.401	0.82	1.23	0.654
	♀	0.94	1.89	0.001 **	0.83	0.83	1.000
Wasting	♂♀	3.58	4.30	0.592	7.22	8.04	0.631
	♂	2.42	4.83	0.190	5.35	4.52	0.677
	♀	4.72	3.77	0.628	9.09	11.57	0.370
Underweight	♂♀	14.80	17.42	0.303	10.93	10.31	0.754
	♂	14.98	17.39	0.506	12.35	13.99	0.593
	♀	14.62	17.45	0.428	9.50	6.61	0.243
Normal weight	♂♀	51.48	58.55	0.027 *	58.70	48.58	0.001 **
	♂	51.64	58.07	0.202	58.48	47.08	0.002 **
	♀	51.32	59.01	0.066	58.93	50.08	0.023 *
Overweight	♂♀	24.89	17.36	0.001 **	21.71	29.56	0.008 **
	♂	24.69	16.81	0.004 **	22.59	30.29	0.016 *
	♀	25.1	17.88	0.014 *	20.83	28.84	0.017 *
Obesity	♂♀	2.86	0.24	0.002 **	0.61	2.42	0.020 *
	♂	2.42	0.48	0.100	0.41	2.88	0.032 *
	♀	3.30	0.00	0.008 **	0.83	2.07	0.254

E: experimental group; C: control group; ♂: boys; ♀: girls; *p*: probability of test statistics, 0 < *p* < 0.001; ** significance at α ≤ 0.01; * significance at α ≤ 0.05.

**Table 12 children-10-00764-t012:** Summary of the results of one-dimensional ANOVA with repeated measurements for fitness components separately for boys and girls.

Variable	Group (E, C)		RM (PRE, POST)		RM*Group	
	*p*	η^2^	*p*	η^2^	*p*	η^2^
Cooper test ♂	0.006 **	0.017	0.000 **	0.644	0.000 **	0.492
Cooper test ♀	0.002 **	0.022	0.000 **	0.509	0.000 **	0.321
Sit-and-reach ♂	0.014 *	0.014	0.000 **	0.476	0.000 **	0.376
Sit-and-reach ♀	0.004 **	0.019	0.000 **	0.541	0.000 **	0.427
Standing broad jump ♂	0.008 **	0.016	0.000 **	0.036	0.001 **	0.023
Standing broad jump ♀	0.006 **	0.0167	0.000 **	0.037	0.002 **	0.022
Handgrip strength ♂	0.000 **	0.111	0.000 **	0.581	0.000 **	0.391
Handgrip strength ♀	0.000 **	0.028	0.000 **	0.445	0.000 **	0.201
Sit-ups ♂	0.000 **	0.034	0.000 **	0.520	0.000 **	0.438
Sit-ups ♀	0.000 **	0.032	0.000 **	0.400	0.000 **	0.315
Bent-arm hang ♂	0.000 **	0.048	0.000 **	0.276	0.000 **	0.213
Bent-arm hang ♀	0.000 **	0.061	0.000 **	0.259	0.000 **	0.204
Agility shuttle run ♂	0.001 **	0.027	0.000 **	0.233	0.000 **	0.151
Agility shuttle run ♀	0.000 **	0.036	0.000 **	0.515	0.000 **	0.355

Variables: fitness traits; ♂: boys; ♀: girls; E: experimental group; C: control group; RM (PRE, POST): repeated measurement—measurement at the beginning and end of the school year; RM*Group: interaction between measurements (PRE and POST) and factor Group (E, C); η^2^: partial eta-square; *p*: probability of test statistics, 0 < *p* < 0.001; ** differences significant at α ≤ 0.01; * differences significant at α ≤ 0.05.

**Table 13 children-10-00764-t013:** Intergroup comparison of the values of the difference between the measurements of boys’ fitness components obtained in I (PRE) and II (POST) of the study—testing with the Student’s *t* test for independent variables.

		Cooper Test I–II	Sit-and-Reach I–II	Standing Broad Jump I–II	Handgrip Strength I–II	Sit-Ups I–II	Bent-Arm Hang I–II	Agility Shuttle Run I–II
	N	X	X	X	X	X	X	X
E	207	−201.626	−3.466	−22.922	−3.888	−3.514	−9.812	2.314
C	243	−31.255	−0.354	−2.523	−0.737	−0.288	−0.847	0.254
*p*		0.000 **	0.000 **	0.001 **	0.000 **	0.000 **	0.000 **	0.000 **

N: number; X: difference of the means obtained in the first and second study; E: experimental group; C: control group; *p*: probability of test statistics, 0 < *p* < 0.001; ** differences significant at α ≤ 0.01.

**Table 14 children-10-00764-t014:** Intergroup comparison of the values of the difference between the measurements of girls’ fitness components obtained in I (PRE) and II (POST) of the study—testing with the Student’s *t* test for independent variables.

		Cooper Test I–II	Sit-and-Reach I–II	Standing Broad Jump I–II	Handgrip Strength I–II	Sit-Ups I–II	Bent-Arm Hang I–II	Agility Shuttle Run I–II
	N	X	X	X	X	X	X	X
E	212	−192.559	−3.673	−22.701	−3.289	−3.289	−6.993	2.144
C	242	−37.293	−0.417	−3.050	−0.950	−0.306	−0.547	0.350
*p*		0.000 **	0.000 **	0.002 **	0.000 **	0.000 **	0.000 **	0.000 **

N: number; X: difference of the means obtained in the first and second study; E: experimental group; C: control group; *p*: probability of test statistics, 0 < *p* < 0.001; ** differences significant at α ≤ 0.01.

## Data Availability

All statistical analyses are in the manuscript. In order to obtain raw data, please contact the authors by e-mail. The data will be shared with all interested parties.

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
