# Peer review of "Assessment of the Impact of Increased Physical Activity on Body Mass and Adipose Tissue Reduction in Overweight and Obese Children"

_children, 2023, doi:10.3390/children10050764_

Round 1
Reviewer 1 Report
Thanks to the authors for the changes made.
The work has been significantly improved.
I propose changing the wording in point 2.1 and Table 1: "Research material". In relation to children, this is an unfortunate term. Maybe "study participants"?
Author Response
Manuscript authors' commentary on the reviewer 1
Comments and Suggestions for Authors
Thanks to the authors for the changes made. The work has been significantly improved.
I propose changing the wording in point 2.1 and Table 1: "Research material". In relation to children, this is an unfortunate term. Maybe "study participants"?
Response
We thank the Reviewer for accepting the changes made to the manuscript. As suggested by the Reviewer, we have changed the title of subsection 2.1 and table 1 in the manuscript from "Research material" to "Study participants".

Reviewer 2 Report
Now, the quality is improved.
Author Response
Manuscript authors' commentary on the reviewer 2
Comments and Suggestions for Authors
Now, the quality is improved.
Response
Once again, we would like to thank the Reviewer for a detailed analysis, constructive suggestions and valuable comments, and for accepting the changes we made to our manuscript after the first round of review.
Reviewer 3 Report
I appreciate your hard work and effort but your manuscript has a lot of issues.
Abstract:
a) The topic and context both emphasize "adipose tissue reduction", but why does the Material and methods section measure "somatic and physical fitness"?
1. Introduction
b) The manuscript says, " A highly important environmental factor influencing obesity is adequately undertaken physical 81 activity. According to scientists, what constitutes a key cause of child and adolescence obesity is absence 82 of physical activity, which is even more crucial than inappropriate nutrition [32]. More and more 83 frequently, physical activity is seen as one of the most effective methods of preventing diseases of 84 civilisation, to which obesity belongs [33, 34, 35, 36](81~84))",it can be assumed that increasing PA will reduce body fat content one is a scientific fact, so what is the innovative point of this paper?
c) The manuscript says,“The results of the interventions used are inconclusive or even 115 contradictory [Al-Khudairy et al 2017, Mead et al 2017, Bilińska and Kryst 2017, Brown et al 2019, Delgado-Floody et al 2019]. ”and I think this piece needs to be expanded upon, i.e., an account of what contradictions led to your research.
2. Material and Methods
d) 2.4 In the section on Statistical methods, LSD and Bonferroni are two different post hoc analysis methods, why are the authors reduced to one?
3. Results
e) Please keep all the numbers in Table 2~13 with three valid digits, and the P value cannot be equal to 0.
Author Response
Manuscript authors' commentary on the reviewer 3
At the outset, we thank the reviewer for detailed analysis, constructive suggestions and valuable comments. We took them all into account and made the necessary corrections to the manuscript. Below we respond to all comments and suggestions:
Comments and Suggestions for Authors
I appreciate your hard work and effort but your manuscript has a lot of issues.
Abstract:
- a) The topic and context both emphasize "adipose tissue reduction", but why does the Material and methods section measure "somatic and physical fitness"?
Response
In the manuscript, a fragment of the abstract text in the "Material and Methods" section has been corrected so that it does not raise any doubts. The new text has been added and highlighted in green.
Abstract: Background: Within the last thirty years, growing child overweight and obesity have been observed as a very concerning phenomenon in most countries worldwide. The research aim was to assess what effect increased physical activity has on reducing body mass and adipose tissue in children between 10-11 years of age as well as to answer the question if physical activity could be considered as a factor preventing child overweight and obesity when maintaining their previous diet and lifestyle. Material and methods: There were 419 sports class primary school pupils in the experimental group who, in addition to four obligatory physical education hours, attended six training hours. The control group comprised 485 children from parallel non-sports classes. All pupils underwent somatic and physical fitness measurements twice: at the beginning and end of school year. In all pupils, height and weight measurements as well as physical fitness measurements were taken twice, at the beginning and end of the school year. Cole's method was used to assess the children's normal body weight. With the help of this method, children with excessive body weight were selected from the entire study group (N-904), who additionally had skinfolds and adipose tissue measured using the BIA electrical bioimpedance method. Obtained results were interpreted using the variance analysis for repeated measurements and LSD test. Results: The number of children with excessive body mass after 10 months of increased physical activity decreased (in the case of overweight: p=0.0014, obesity: p=0.0022), as did their skinfolds (p≤0.001) and body fat (p≤0.001), while their physical fitness considerably improved compared to the control group. Conclusions: The introduction of increased physical activity in the experimental group children, when maintaining their existing diet and lifestyle, contributed to reducing their obesity and overweight and, all the same, proved to be an effective factor in the process of decreasing their excessive body mass.
- Introduction
- b) The manuscript says, " A highly important environmental factor influencing obesity is adequately undertaken physical 81 activity. According to scientists, what constitutes a key cause of child and adolescence obesity is absence 82 of physical activity, which is even more crucial than inappropriate nutrition [32]. More and more 83 frequently, physical activity is seen as one of the most effective methods of preventing diseases of 84 civilisation, to which obesity belongs [33, 34, 35, 36](81~84))",it can be assumed that increasing PA will reduce body fat content one is a scientific fact, so what is the innovative point of this paper?
Response
The fact that increased physical activity significantly improves the body composition of the obese population has been examined and corroborated. Our study involved two groups of children who did not differ significantly at the beginning of the school year in either body weight and height, physical fitness, diet or lifestyle. The only environmental factor that was changed was the introduction of an additional 6 hours per week of physical activity in one group. The other factors were not changed, which was verified by a questionnaire completed by parents at the beginning and end of the school year. The aim of our study was to answer the question: whether increased physical activity without changes in diet and lifestyle by 6 additional hours of physical education per week for the whole school year, i.e. 10 months, would lead to a reduction in body weight and body fat of overweight and obese children.
The results of our research provided a positive answer to the question raised and they have confirmed that is it is necessary to increase children’s physical activity. Our study conclusions form a strong argument for the prevention of the development of child overweight and obesity, and they indicate that counteracting obesity is a crucial task of physical education. For, as it transpires from our study, the introduction of only the factor of increased physical activity (which, in our case, was an additional six hours of physical education continuously in the whole school year) is enough as a minimum but effective way of preventing child obesity. This is what we consider as significant contribution to research on combating child obesity that has been conducted so far.
It should be conceded today that obesity prevention constitutes a major social issue and an important task for physical education which should target current and future health-related needs in its curricula. Since there is an insufficient number of physical education lessons in schools, it is mainly ministry of education institutions which our research addresses.
A specific novelty of our research is:
- The isolation of the physical activity factor of precise intensity (additional six hours of weekly physical education lessons continuously in the whole school year) when maintaining the existing diet and lifestyle.
Advantages of our research:
- The introduction of a strictly controlled intervention, directed by a teacher or an instructor for 10 school year months (children with absence exceeding one week were excluded from the study);
- Appropriate choice of research material: groups included in our research comprised over 400 children residing on one town, characterised by similar environmental factors influencing the prevalence of overweight and obesity (ways of how children spend their free time, how they engage in organised forms of physical activity, family physical activity, and eating fast-food meals) and a similar socio-economic status, and they differed only with regard to the size of intervention (that is the number of physical education hours).
- c) The manuscript says,“The results of the interventions used are inconclusive or even 115 contradictory [Al-Khudairy et al 2017, Mead et al 2017, Bilińska and Kryst 2017, Brown et al 2019, Delgado-Floody et al 2019]. ”and I think this piece needs to be expanded upon, i.e., an account of what contradictions led to your research.
Response
Although the role of physical activity in obesity prevention is undisputed, its use does not always have a positive effect. Results of interventions made in recent years are not conclusive, they are even contradictory, which is shown in results of research conducted so far which we presented in our manuscript, in chapter “Discussion.” Quote: ”Research conducted by Januszek–Trzciąkowska and co-authors [2014] did not show any statistically significant connection between a physical activity level and obesity in 7–9-year-olds…. In the course of their study among 10–12-year-old children… . Czajka and co-authors [2012] found an insignificant relationship between a BMI value and physical activity, which was defined as physical exertion lasting at least 20 minutes a day. In a cross-sectional study performed in Great Britain among children between 6–8 years of age, found that a higher level of physical activity was associated with a lower risk of overweight and obesity, but only in boys [Basterfield et al. 2014]. …Olson and co-authors [2014] revealed at the level of statistical tendency that there was association between a number of hours spent on active play or sport and a risk of overweight and obesity in children aged 6–11 (p = 0.06). …Koca and co-authors [2017] observed higher BMI values in the group of physically active children. Whereas Janssen and LeBlanc [ ] who in their systematic review analysed thirty-one observational studies concerning physical activity and excessive body mass among children showed that most research pointed to a poor or moderate connection between an elevated level of physical activity and lower risk of overweight and obesity. ….Research conducted by Bilińska and Kryst [2017] aimed at determine the effects on overweight/obesity prevalence of the primary-school-based intervention program. After one year of extra physical activities and engagement in health-oriented education program, the risk of being overweight/obese was not reduced in children in the experimental group. …The results of a review of strategies to control childhood obesity based on 105 scientific articles showed that various projects produced sometimes controversial results [Kelishadi and Azizi-Soleiman 2010]. …Mead et al. [2017] reviewed the studies for evaluation the effects of diet, physical activity and behavioural interventions (behaviour-changing interventions) for the treatment of overweight or obese children aged 6 to 11 years. The quality of the evidence was low or very low and the results were heterogeneity. They concluded that more research on interventions is needed.…” Brown et al [2019] assessed the effectiveness of a range of interventions (154 RCT) that include diet or physical activity components, or both. The effectiveness of the intervention based only on physical activity depended on the age of the children. The results of studies by other authors indicate a positive effect of physical activity on the incidence of obesity in children [Charzewska et al. 2006, Baran 2018, Rutkowski et al. 2019, Ługowska et al. 2022].
This ambiguity in the findings of the literature is due to the lack of comparable criteria for the use of physical activity. The type of activity used, the frequency and duration of training, and the possible correlation of physical activity with other environmental factors influencing obesity (e.g. diet, sleep, sedentary lifestyle) and the size and age of the study group are important. The discrepancies in the literature regarding the effectiveness against obesity have confirmed the need for studies in which physical activity is strictly defined and monitored. The study we conducted was exactly that. An additional six physical education lessons per week conducted under the guidance of a trainer were introduced for a period of 10 months. The only factor changed was the increase in physical activity, all other factors remained the same.
- Material and Methods
- d) 2.4 In the section on Statistical methods, LSD and Bonferroni are two different post hoc analysis methods, why are the authors reduced to one?
Response
The Least Significant Difference (LSD) test of multiple pairwise comparisons is equivalent to multiple separate t-tests between all pairs of groups. The disadvantage of this method is that it does not adjust the observed significance level for multiple comparisons. Therefore, we additionally used the Bonferroni test, which is calculated in the same way as the LSD test, but takes a correction for the number of comparisons made. In our study, we made 6 comparisons (between E - PRE and E -POST, E -PRE and C - PRE, E-PRE and C-POST, E-POST and C - PRE, E -POST and C -POST, C-PRE and C- POST), therefore a significance level of 0.05 had to be divided by 6 (=0.0083). Thus, only a significance result approximately less than 0.008 is considered statistically significant.
In the "Material and Methods" section, in subsection “Statistical methods” we have corrected the text to make it clearer. The legends of tables 3-6 showing the significance of differences between groups (LSD test results) include information about the application of the Bonferroni correction.
- Results
- e) Please keep all the numbers in Table 2~13 with three valid digits, and the P value cannot be equal to 0.
Response
Following the Reviewer’s suggestion, we have applied corrections to Tables 2 to 13. We have rounded the numbers contained therein to three decimal places. Therefore, in Tables 3, 5, and 6 we have written zero, because in actuality it p was smaller than 0.0001

Reviewer 4 Report
This is an important topic and incorporating the research design into a real life setting spread over an entire school year adds external validity. There are a number of improvements that could be made to the manuscript to make it more reader friendly and add clarity.
- Title is long and awkward. Perhaps "Efficacy" could replace "Effectiveness assessment"
- Methods could be explained better. The exact protocols should be noted. Did the same assessor do both pre and post measures? Were they experienced? Certified? Were assessments done at the same time of day pre and post (i.e. BIA can change over the course of a day due to hydration levels). Was diet monitored (plausible that initiation of an exercise program motivated individuals to also improve their diet).
- In addition to explaining the methods better, the authors could acknowledge limitations to the study when proper controls were not feasible.
Despite these shortcomings the study has merit and contributes to the field.
Author Response
Manuscript authors' commentary on the reviewer 4
At the outset, we thank the reviewer for detailed analysis, constructive suggestions and valuable comments. We took them all into account and made the necessary corrections to the manuscript. Below we respond to all comments and suggestions:
Comments and Suggestions for Authors
This is an important topic and incorporating the research design into a real life setting spread over an entire school year adds external validity. There are a number of improvements that could be made to the manuscript to make it more reader friendly and add clarity.
- Title is long and awkward. Perhaps "Efficacy" could replace "Effectiveness assessment"
Response
The title of the manuscript is indeed long, but we wanted it to fully inform the reader what the manuscript will be about. Following the advice of the Reviewer, we shortened the title and it currently reads: „Assessment of the impact of increased physical activity on body mass and adipose tissue reduction in overweight and obese children”.
- Methods could be explained better. The exact protocols should be noted. Did the same assessor do both pre and post measures? Were they experienced? Certified? Were assessments done at the same time of day pre and post (i.e. BIA can change over the course of a day due to hydration levels). Was diet monitored (plausible that initiation of an exercise program motivated individuals to also improve their diet).
Response
All measurements (PRE and POST) were performed by the same person, an anthropologist with 10 years of experience, who was a blind investigator (did not know about the children's belonging to the experimental or control group. Somatic measurements, i.e., height and weight, as well as fat folds and body fat levels, were taken in the morning always at the same time. Children were fasted, after these measurements they received a light breakfast at school and were subjected to fitness measurements.
All measurements were taken in line with binding procedures in a room next to the swimming pool where pupils attended obligatory swimming classes. Physical fitness tests were the only activities conducted in gymnasiums of individual schools.
To evaluate environmental factors conditioning the prevalence of overweight and obesity in the population of children under study, an author’s questionnaire was used which was completed by the children's parents. It was made sure that the children in both groups (experimental and control) were exposed to similar environmental factors with regard to how a child spent its leisure time, what organised forms of physical activity and family physical activity it took part in and what kind of fast-food meals it ate.
Children whose diet changed for any reason during the observed period were excluded from the study (based on information from the survey filled out by parents).
The reviewer's suggestions have been incorporated into the manuscript.
- In addition to explaining the methods better, the authors could acknowledge limitations to the study when proper controls were not feasible.
Despite these shortcomings the study has merit and contributes to the field.
Response
The sample size limited the analysis in determining various factors regarding physical activity.
Although all primary schools (there were 10) in the city of Leszno (population 60,000) were included in the study, a larger number of children surveyed would have provided greater opportunities to analyse the various factors that are components of physical activity (type of activity, training time units) and to obtain more detailed results. A larger study sample size would allow for categorisation of children in terms of quantity and quality of physical activity practice. Future studies could be supplemented with further age groups of children.

Round 2
Reviewer 3 Report
I think that it can be accept in present form.